# Oral Microbiota and Tumor—A New Perspective of Tumor Pathogenesis

**DOI:** 10.3390/microorganisms10112206

**Published:** 2022-11-08

**Authors:** Simin Li, Mingxin He, Yumeng Lei, Yang Liu, Xinquan Li, Xiaochen Xiang, Qingming Wu, Qiang Wang

**Affiliations:** 1Institute of Infection, Immunology and Tumor Microenvironment, Hubei Province Key Laboratory of Occupational Hazard Identification and Control, Medical College, Wuhan University of Science and Technology, Wuhan 430065, China; 2Wuhan Asia General Hospital Affiliated to Wuhan University of Science and Technology, Wuhan 430065, China

**Keywords:** oral microbiota, tumor pathogenesis, immune system, carcinogenesis, tumor diagnosis

## Abstract

Microorganisms have long been known to play key roles in the initiation and development of tumors. The oral microbiota and tumorigenesis have been linked in epidemiological research relating to molecular pathology. Notably, some bacteria can impact distal tumors by their gastrointestinal or blood-borne transmission under pathological circumstances. Certain bacteria drive tumorigenesis and progression through direct or indirect immune system actions. This review systemically discusses the recent advances in the field of oral microecology and tumor, including the oncogenic role of oral microbial abnormalities and various potential carcinogenesis mechanisms (excessive inflammatory response, host immunosuppression, anti-apoptotic activity, and carcinogen secretion) to introduce future directions for effective tumor prevention.

## 1. Introduction

In recent years, increasing studies have shown a direct link between the gut microbiota and the development of various human diseases. They may reduce the effectiveness of drugs [1], affect the development and behavior of the host’s brain [2], and play an important role in tumor occurrence, progression, and treatment [3]. The oral microbiota is the second most complex microbiota in the human body, besides the gut microbiota. It can also impact health and has been associated with oral inflammation, local and systemic diseases, and cancer. Disturbances in the diversity and ratios of species within the oral microbiota lead to dysbiosis and an increased risk of associated diseases, including periodontitis [4].

The common oral microbiota, also known as symbiotic bacteria, causes no discomfort to the body nor is it beneficial. *Streptococcus dentisani* is a common inhabitant of the oral microbiota, worldwide, which inhibits the growth of pathogens through the production of bacteriocins [5]. Members that can cause infection in the host are called oral pathogens (e.g., *Prevotella, Selenomonas,* and *Atopobium*) [6]. Selenomonas may cause serious human diseases including bacteremia [7], and they are associated with asthma severity [8]. In addition, certain members, called opportunistic pathogens, are not pathogenic under normal conditions but can cause disease under specific conditions such as by decreasing the host’s immune defenses or dysbiosis. Several oral taxa, especially *Porphyromonas gingivalis (P. gingivalis)* and *Fusobacterium nucleatum (F. nucleatum)*, have been shown to have carcinogenic potential through several diverse mechanisms [9]. Accordingly, this review is focused on elucidating the relationship between oral microbiota and tumors (Figure 1).

## 2. Systemic Involvement of Oral Microecology

An eclectic and diverse assemblage of microbiota inhabits different sites within the oral cavity, and more than 700 bacterial species or phylotypes have been identified [10,11]. The oral cavity has several distinct microbial habitats, including the subgingival plaque, supragingival plaque, buccal mucosa, keratinized gingiva, saliva, hard palate, tongue, and tonsil [12]. The previous study analyzed the tongue dorsum, hard palate, and several different parts of the oral cavity and saliva with or without rinsing in 20 healthy subjects, and it demonstrated that no significant differences were seen between the study participants [13]. Therefore, the saliva microbiota and oral wash microbiota, including other bacterial components in the oral cavity, show long-term stability, which can reflect the oral microbiota to a certain extent [14]. The microorganisms that are detected in saliva may be diagnostic markers for many diseases, so saliva can provide a ‘window’ to explore personal health [15]. Likewise, an oral wash can be used to provide a sample to analyze the oral microbiota characteristics [16].

Oral pathogens spread to distant organs through oral and maxillofacial blood circulation or participate in systemic circulation through the gastrointestinal tract [17]. The imbalance of the oral microecology is not only associated with oral diseases, but it is also confirmed to be related to the occurrence and development of several systemic diseases such as cardiovascular diseases [18], respiratory diseases [19], immune system diseases [20], metabolic diseases [21], and even nervous system diseases [22]. At the same time, oral pathogens play an important role in cancer development such as in colorectal cancer [23]. Subsequent studies should provide a deeper and more comprehensive understanding of the oral microecology based on the changes in the oral microbiota (including changes in abundance and diversity) in diseased and healthy states.

## 3. Dysregulated Oral Microbiota Poses a Challenge to the Immune System

Oral microbiota maintains a dynamic symbiotic relationship of dependence and constraint with the host’s immunity. It is well established that microbial communities can form a natural barrier with colonization resistance (including the production of adhesions, lectins, amylases, mucins, etc.) which can resist the colonization and establishment of exogenous microorganisms [24]. Willems et al. argued that *Candida albicans* (*C. albicans*) might decrease the cariogenic potential of a Gram-positive bacterium such as *Streptococcus pyogenes* by increasing the pH within mixed biofilms [25]. The common microbiota members, such as *Staphylococcus epidermidis*, can also inhibit the formation of Staphylococcus aureus biofilm [26].

An interspecific interaction may affect their ability to adhere to and invade epithelial cells. For example, *Streptococcus sanguinis* can affect the biofilm formation and the expression of the pathogenic genes spaP, gtfB, and gbpB in *Streptococcus mutans* [27]. *Treponema denticola* can form pores within the biofilms and facilitate the flow of nutrients to pathogenic bacteria such as *P. gingivalis* [28]. Overall, the invading pathogens can affect the mechanism of the epithelial barrier function by manipulating the barrier-associated genes or proteins to adhere and subsequently, internalize or by directly disrupting the connections to provide access to the underlying tissue [29,30]. Representative pathogens include *F. nucleatum, Bacteroides forsythus*, and *Aggregatibacter actinomycetemcomitans (A. actinomycetemcomitans)*. These members possess potent virulence factors, including lipopolysaccharide (LPS) and metalloproteinases (MMPs), which facilitate their entry into the underlying tissues and stimulate the release of proinflammatory cytokines [31].

The variability in the relative abundance, structural composition and space of different bacterial species may, to some extent, transform highly organized biofilm communities from a fragile symbiotic state to an imbalanced one. In the case of a disbalance (dysbiosis), the leakage of these microbial products negatively affects the immune system, including by activating the extracellular matrix degradation pathway and destroying the immune-related signal pathways, thereby contributing to a chronic proinflammatory state or cancer [32,33,34]. The selection and enrichment of pathogenic bacteria, such as *K. aerogenesa*, potentially enhance the local or distal destructive inflammatory responses by activating inflammasome-mediated IL-1 signaling in macrophages [35]. Infection by opportunistic pathogens such as *P*. *gingivalis* and *A. actinomycetemcomitans* can induce a differentiated production of cytokines and bacterial products, including IL-1, IL-6, IL-8, and TNF-α [36].

Restoring the oral microecology from an imbalanced state to a normal state for oral diseases and even oral mucositis that is induced by chemoradiotherapy has not yet been reported. There is a distinctive predominant bacterial flora of the healthy oral cavity that is highly diverse and is different from that of oral diseases. Aas J A et al. detected 141 different representative bacterial taxa in the oral cavity of healthy people [10]. However, when these bacteria lose their colonization advantage, the oral microbiome transforms into a dysbiotic state, including a PH imbalance and symbiotic biofilms dysbiosis, which may accelerate the pathological processes of numerous diseases by serving as a storeroom for opportunistic pathogens [37]. More importantly, studies have shown that invasive pathogens can use specific pathways such as the T6SS gene cluster to protect their ecological niche and maintain colonization [38]. However, there is still a lack of research on how to recover dysfunctional oral microbiota.

## 4. Is the Oral Microbiota a Cause or a Consequence of Tumorigenesis?

The imbalance, colonization, and translocation of the oral microbiota are likely to be important influences on the progression of tumors in situ or distantly [39,40]. Oral anaerobes are potentially pathogenic, particularly *F. nucleatum and P. gingivalis*, which are closely associated with various types of tumors [41,42]. Some aerobic bacteria, such as *Parvimonas*, are also associated with tumorigenesis [43]. An imbalance in the overall bacterial community (change of diversity metrics) is involved in the development of tumors, which is mainly based on chronic inflammation and immunosuppressive. The changes in the abundance of individual bacterial species, particularly *Peptostreptococcus, Prevotella,* and *Parvimonas,* can also induce chronic inflammation. These bacteria cause the up-regulation of numerous cytokines (IL-1β, IL-6, IL-17, IL-23, TNF-α, etc.) and inflammatory mediators (matrix metalloproteinases MMP-8 and MMP-9) [44,45]. Identifying the relationship between the tumors and the oral microecology in a complex biological environment is essential and urgently needs to be achieved.

In addition, tumor progression may be driven by interactions between the host’s immune system and the microbial metabolites (in the case of lactate and dietary tryptophan metabolites, for example). Lactate can recruit and induce immunosuppressive cell types, such as regulatory T cells, tumor-associated macrophages, and myeloid-derived suppressor cells, thereby suppressing the anti-tumor immune responses [46]. Metabolites of dietary tryptophan activate the aryl hydrocarbon receptor in myeloid cells, promoting an immune-suppressive tumor microenvironment and facilitating pancreatic ductal adenocarcinoma growth [47].

Furthermore, there are multiple ways for cancer cells to evade detection and destruction by the immune system during tumor development, namely immune evasion. Carcinogens produced by microbiota contribute to the immune escape of tumors by interfering with the recognition ability of the immune cells and depleting T cells [48]. Certain oral microbiota and their metabolites have tumor-promoting properties, while the tumor microenvironment contributes to the colonization of the microbiota. In conclusion, oral microbiota has attracted reasonable attention for its involvement in tumor progression. More studies are needed to provide evidence to support that oral microbial therapy could offer a striking clinical benefit for cancer patients.

### 4.1. Oral Cancer and Oral Microbiota

Oral cancer is one of the most common malignancies of the head and neck, of which more than 90% are oral squamous cell carcinomas (OSCC) [49]. Poor oral hygiene and chronic inflammation alter specific microbiota, which together with their metabolites, are risk factors for the development and progression of oral cancer [50]. In particular, biological dysbiosis in the oral cavity of patients with periodontitis may lead to potentially malignant mucosal lesions in the oral cavity, thereby promoting the development and progression of cancer [51,52].

Researchers have used a bioinformatics analysis to note significant differences in the abundance and diversity of oral microbiota in the cancer patient group compared to those in the healthy group [53]. In terms of oral microbiota distribution, the ratios of dominant bacteria *Staphylococcus* and *Rothia* were significantly higher in the cancer group than they were in the control group [54]. Furthermore, the saliva having direct contact with the oral cancer lesions makes it a more specific and potentially sensitive screening tool, whereas more than 100 salivary biomarkers (DNA, RNA, mRNA, and protein markers) have already been identified, including cytokines (IL-8, IL-1b, TNF-α), defensin-1, P53, Cyfra 21-1, tissue polypeptide-specific antigen, dual-specificity phosphatase, spermidine/spermineN1-acetyltransferase, profilin, cofilin-1, transferrin, and many more [49]. These studies suggest that colonization by specific microbiota might be important for the development and prognosis of oral cancer (Table 1).

The relationship between some single species-specific microorganisms such as pathogens and cancer has been investigated. For instance, Staphylococcus aureus, a pathogen which is prevalent in oral cancer, can also activate the COX-2/PGE 2 pathway in human oral keratinocytes (HOK) cells and play a role in tumor progression [58]. Polymicrobial interactions of *C. albicans* with other members of the oral microbiome have been reported to enhance the malignant phenotype of oral cancer cells, such as the attachment to extracellular matrix molecules (ECM) and epithelial–mesenchymal transition (EMT) [59]. One possible mechanism is that *C. albicans* could create ideal conditions for pathogens to survive and colonize, and this co-infection might lead to more severe pathogenicity and drug resistance [60]. On the other hand, The presence of *C. albicans* increased the production of extracellular polysaccharides (EPS) and the coexistence of EPS with Candida albicans induced the expression of Streptococcus pyogenes virulence genes (e.g., gtfB, fabM) [61,62].

The metabolic propensity of the oral microbiota further reveals the mechanisms of oral carcinogenesis. The most abundant microbial metabolic pathways in the tumor tissues of oral cancer patients were those related to fatty-acid biosynthesis, carbon metabolism, and amino acid metabolism [53]. For instance, a *Pseudomonas gingivalis* infection significantly increases the level of free fatty acids in the tongue and serum of mice, thereby altering the fatty acid profile, exacerbating the disruption of fatty acid metabolism, and ultimately promoting the development of oral cancer [63].

### 4.2. Gastric Cancer and Oral Microbiota

Gastric cancer (GC) is a major health problem in many countries with high incidence and mortality rates [64]. Other than the host factors (genetics and age), the environmental factors including microbial infections have been shown to contribute to gastric carcinogenesis [65]. In recent years, studies on the relevance of oral microecology to GC have been increasing, and these have revealed that the oral microbiota may play an important role in GC and precancerous stages (Table 2).

As a note of interest, the similarities between the gastric and oral microecology can be observed in cohort studies [66,67,69]. Oral symbionts including *Parvimonas, Eikenella, Prevotella-2, Slackia, Selenomonas, Bergeyella,* and *Capnocytophaga* have been identified with high relative abundances in the gastric mucosa of GC patients [65]. The translocation and diffusion of the oral microbiota may induce the occurrence and progression of GC through distal effects. As an early warning and a preventive indicator of precancerous lesions and GC, *Helicobacter pylori (H. pylori)* is not only present in the stomach of patients with GC, but it can also be detected in the oral cavity [68,74]. *H. pylori* could affect the retention and colonization of the oral microbiota by inducing the production of salivary mucin MUC5 B and MUC7 [75].

One possible mechanism for the involvement of the oral microbiota in carcinogenesis is the enrichment of pro-inflammatory oral bacterial species. GC patients’ saliva and gastric mucosa are enriched with *Corynebacterium*, which is increasingly reported as an emerging opportunistic pathogen in cancer, hematological malignancy, and in critically ill patients [66]. *Fusobacterium* species have also received a lot of attention due to their pro-inflammatory nature, with TLR4 and autophagy playing very important roles in the inflammation that they induce [73]. Furthermore, *Campylobacter concisus* which is enriched in the tongues of patients with gastritis could induce the expression of cytokines and chemokines such as TNF, IL-1β, and IL-10 [71].

Another possible mechanism for the involvement of the oral microbiota in carcinogenesis is the abnormal enrichment of bacterial metabolites, which are direct-acting carcinogens, and these could persist stably in the gastrointestinal tract. Studies have shown that acid-producing bacteria and carbohydrate metabolic pathways were more abundant in patients with GC, which results in the significant formation of short-chain fatty acids (SCFA) and lactate [73]. Lactate, as a source of energy for tumor cells, was able to induce the cellular glycolytic pathways, increase the ATP supply, enhance inflammation, and activate tumor angiogenesis [76,77]. However, it is not clear why in some cases these bacterial metabolites present inhibitory effects in inflammatory responses and cancer development [78,79,80], whereas in others, they have the opposite effect [81,82]. The fact that different concentrations of them are enriched in the host and different molecular mechanisms are activated on various types of cells might partly explain these contradictions. Meanwhile, the salivary microbiota of GC patients is involved in the upregulation of the isoleucine and valine biosynthetic pathways [66]. Purines could regulate the immune cell responses and cytokine release, contributing to the tumor microenvironment [68]. In addition, a decreased abundance of hexitol metabolism-related microbial gene families and metabolic pathways and an increased abundance of microbial TCA cycles II and VII were associated with an increased GC risk [72]. The oral microbiota including *Streptococcus salivarius, Streptococcus retardans,* and *Streptococcus mucosus* may produce alcohol dehydrogenase (ADH), which metabolizes ethanol into carcinogenic acetaldehyde [83].

Of great attention is the nitrosamine hypothesis of gastric carcinogenesis [84,85]. Studies have shown that when they are compared with those of the non-cancer group, the metabolic enzymes related to denitrification, including nitrate reductase and nitrous oxide reductase, were enriched in the gastric microbiota of the cancer group. Most microorganisms in the stomach are considered to come from the external environment, particularly the oral cavity. The oral microbiota can enter the stomach through saliva and cause pathological changes in the corresponding areas [86]. The accumulation of nitrogenous compounds such as nitrates and nitrites in the stomach increases the risk of GC and promotes the malignant transformation of cells in the stomach [68]. Studies have also revealed a low relative abundance of *Haemophilus parainfluenzae* and *Nitrospirae* in GC patients. These are both nitrate-reducing bacteria that convert nitrate to nitrite or nitric oxide (NO) for their absorption through the oral vasculature or to be swallowed into the gastrointestinal system. The accumulation of N-nitroso compounds in the gastrointestinal tract may increase the cancer risk [66]. These findings support that there are complex causal relationships between the salivary microbiota and the gastric tumor formation.

### 4.3. Colorectal Cancer and Oral Microbiota

Colorectal cancer (CRC) is a common malignant tumor of the gastrointestinal tract with an increasing incidence in the last decade [87,88,89]. The causes of CRC are complex and include a variety of factors such as high-fat and low-fiber dietary habits and obesity [90]. In recent years, mounting evidence supports the link between the oral microbiota and CRC (Table 3), with persistent periodontal inflammation exacerbating the development of CRC [91].

*F. nucleatum* is a well-known pro-inflammatory, aggressive anaerobic oral pathogen that is capable of participating in the progression of CRC [44,101]. The outer membrane protein Fap2 is known to mediate bacterial enrichment in CRC by binding to tumor-expressed Gal-GalNAc. The adhesin FadA promotes E-cadherin/β-catenin signaling and prevents immune attacks by binding its bacterial protein Fap2 to the inhibitory immune receptor TIGIT on NK and T cells [98,102,103]. *F. nucleatum* host-cell binding and invasion induces the IL-8 and CXCL1 secretion that drives CRC cell migration [9]. Furthermore, *F. nucleatum* can also bind to CEACAM1 to evade an immune attack and generate a pro-inflammatory microenvironment by recruiting tumor-infiltrating immune cells [104,105]. Hong et al. showed that *F. nucleatum* targets lncRNA ENO1-IT1 to promote glycolysis and oncogenesis in colorectal cancer [106]. Therefore, targeted therapies such as targeting the ENO1 pathway may have implications for the treatment of CRC patients with elevated *F. nucleatum* levels. These results raise the possibility that the oral microbiome may play an important role in CRC etiology.

Notably, Carolina et al. found a high similarity between the *F. nucleatum* strains that were found in tumor tissues and saliva from CRC patients, highlighting the oral origin of them and strengthening the hypothesis of an oral-driven dysbiosis of intestinal ecology in CRC [107]. Subsequently, by analysis of the stool samples from 252 healthy and advanced CRC subjects, a significant increase in the relative abundance of the species of oral-derived microbiota, including but not limited to *Peptostreptococcus stomatis, Parvimonas micra, Gemella morbillorum,* and *F. nucleatum*, was found in the gut of CRC patients [95]. To determine the mechanism by which oral bacteria diffuse into the gut, Bolei Li et al. observed the bacterial colonization of the gastrointestinal tract by transplanting human saliva into germ-free (GF) mice [108]. On these bases, the mechanisms of transfer of the oral microbiota in the pathogenesis of CRC have been proposed: (i) The transmission of the oral bacteria to the intestinal environment through continuous swallowing. The oral bacteria that are capable of resisting the harsh acidic gastric environment can maintain their viability through this pathway [107]. The most representative one is *P. gingivalis*, a characteristic contributing to its migration to the distal tissues, thereby altering the composition and functional capacity of the residual microbiota in the pathological gut [44]. (ii) The spread of the oral bacteria into the intestinal environment via the bloodstream (bacteremia) or a lymphatic route. During oral cleaning, a tooth extraction and bacteremia, the gingival epithelium breaks down or becomes more permeable, whereby the oral microbiota may spread directly to the distal area through the bloodstream [109]. During chronic periodontitis, the circulatory system appears to be the most efficient way for *F. nucleatum* to reach the colorectum [110,111]. (iii) The altered intestinal environment encourages the invasion and colonization of the oral bacteria in the gut [112]. The invasion of oral microbiota inevitably results in the instability of the commensal microbiota and facilitates the colonization of it by oral opportunistic pathogens [44]. Of these, *Prevotella intermedia* was thought to cooperate with other oral pathogens to colonize the colon and persist as a population, forming an inflammatory microenvironment that may promote the development of CRC [97].

The oral microbiota reaches the intestinal mucosal sites and negatively affects CRC through immunosuppressive and toxic effects. On the one hand, the intestinal translocation of the oral bacteria may cause intestinal mucosal damage. The metabolic profile of the oral bacteria in the colon is characterized by the glycolytic and proteolytic metabolism, which is capable of degrading mucin and extracellular matrix in the colon, leading to the infiltration of the mucus layer and the invasion of the mucosa through disruption of the epithelial junctions [107]. In addition, the oral members of the gut microbiota were also able to metabolize tryptophan into various derivatives (tryptamine, indole, and fecal odorant) that modulate the immune response of the colonic epithelium by binding to the aryl hydrocarbon receptor (AhR) [44]. Among them, specific bacterial species with tumor-promoting properties may activate the inflammasome and NF-κB cascade pathway and induce DNA damage in the intestinal epithelium, thereby promoting CRC [111]. On the other hand, various carcinogenic metabolites, reactive oxygen species, and polyamines which are synthesized by members of the oral microbiota may also be causative factors [107]. For example, oral-derived bacteria such as *A. actinomycetemcomitans, F. nucleatum, P. intermedia,* and *P. gingivalis* could produce volatile sulfur compounds (VSCs), including hydrogen sulfide (H2S), which have a toxic and inflammatory potential even at low concentrations [113]. *Peptostreptococcus anaerobius* can also increase the cholesterol production and the cell proliferation in a reactive oxygen-dependent manner, thereby promoting colon cancer formation [114].

The oral microbiota of CRC patients is unique and predictive, so testing the oral microbiota may provide new directions for the prevention and screening of CRC [115]. Burkhardt et al. used a combination of oral and fecal microbiota to test for CRC with higher specificity (95%) and sensitivity (88%) than the commonly used Fecal Immuno-Test (FIT) can, suggesting that the information from the oral microbiota could potentially improve the performance of current diagnostic tests [98]. Particularly promising is its high sensitivity (88%) for detecting colorectal adenomas because of the importance of the early detection of colonic disease for cancer prognosis and treatment [98].

### 4.4. Pancreatic Cancer and Oral Microbiota

The incidence and mortality of pancreatic cancer (PC) (including non-ductal tumors, pancreatic ductal adenocarcinoma (PDAC) and its classical precursor lesions) have been on the rise, worldwide [116,117]. It has been found in many epidemiological surveys that poor oral health can lead to an increased risk of PC [17]. A meta-analysis showed an overall positive association between periodontitis and pancreatic cancer, even after adjusting for the common risk factors [118].

The mechanism of the link between oral disease and PC is unclear, but it may be related to alterations in the oral microbiome. Several cohort studies have reported associations among oral health, periodontitis (PD), and PC risk, and increased levels of antibodies associated with oral microbiota are associated with a higher risk of PC (Table 4). Both cohort studies by Céline Tiffon et al. and Ai-Lin Wei et al. showed that the saliva microbiota can distinguish between normal and cancer patients as well as being a promising non-invasive diagnostic tool for PDAC [119,120]. Similarly, Lu et al. demonstrated a significant increase in the microbiome diversity in the tongue coat of PHC patients [121].

Interestingly, the conventional view is that the pancreas is a sterile organ, yet in the cancer setting, there is a distinctly specific microbiota in the pancreatic capsule fluid samples which overlaps with the oral microbiota [126,127]. Gaiser et al. observed the coexistence and enrichment of the oral bacterial taxa including F. nucleatum and Granulicatella adiacens in the cyst fluid of IPMN with high-grade dysplasia [118]. In addition, the study of Alkharaan et al. showed that circulating antibodies against commensal oral bacteria appeared to be elevated in the PC patients [125]. Therefore, the oral microbiota may metastasize and colonize the PC through blood circulation, causing alterations to the tumor microenvironment [17]. The researchers speculate that, on the one hand, the hypoxic and immunosuppressive nature of the pancreatic tumor microenvironment supports the preferential growth of oral anaerobic bacteria [118]. On the other hand, the similarity in function that is shared between the pancreas and the salivary glands may also create a biological environment that attracts similar microorganisms [119].

The systemic transmission of dysregulated oral microorganisms and their toxins may modulate the signaling pathways and metabolic pathways, which contributes to the risk of PC. Gnanasekaran et al. demonstrated that intracellular *P. gingivalis* promotes the tumorigenic behavior of pancreatic cancer cells by the activation of the Akt signaling cascade [128]. Bacterial metabolites (short-chain fatty acids, secondary bile acids, polyamines, indole derivatives, etc.) could also play an important role in microbiome-driven pancreatic adenocarcinoma [129]. Thus, the oral microbiota may serve as a potential biomarker for pancreatic cancer detection to identify high-risk individuals for pancreatic cancer initiation, progression, or poor prognosis and improve our understanding of its pathogenesis [130].

### 4.5. Lung Cancer and Oral Microbiota

Lung cancer is one of the malignant tumors with the fastest increase in morbidity and mortality, posing a great threat to human health and life [131]. It is well known that smoking is a major risk factor for lung cancer. Studies have shown that the alpha diversity in smokers in the buccal mucosa is significantly lower than it is in nonsmokers, and the relative abundance of different taxa was significantly different due to their smoking status [132].

Recently, it has been found that the oral microbiota has a potential role in lung cancer (Table 5). Mi Young Lim et al. compared the oral microbiota of patients with lung cancer to a healthy control group, and they found that there were significant differences in the structure of the oral microbiota; the Shannon diversity index was significantly lower [133]. The risk of lung cancer was negatively correlated with the α diversity of the oral microbiota, and the abundance of some specific taxa can be used as potential biomarkers [134]. Another large prospective study on the oral microbiome and lung cancer also demonstrated that multiple oral microbial measures were associated with lung cancer risk, especially in squamous cell carcinoma and smokers [135].

Moreover, common oral microbiota members can usually be detected in the lower respiratory tract of lung cancer patients. By investigating the infection of *P. gingivalis* in lung cancer tissues, Liu et al. proposed that long-term smoking and alcohol consumption would cause a bad oral environment and increase the risk of *P. gingivalis* infection, and then, *P. gingivalis* infection will promote the malignant progression of lung cancer [137]. The enrichment of oral bacteria *Veillonella parvula* may relate to the up-regulation of the carcinogenic pathways (such as IL-17, PI3K, MAPK, and ERK pathways) and the activation of the checkpoint inhibitor markers, thereby affecting tumor progression and prognosis [136]. More research should be carried out to explore the possibility of this potential targeted therapy.

### 4.6. Breast Cancer and Oral Microbiota

The incidence of breast cancer (BC) is increasing year after year, ranking first in the incidence of female cancer [138]. A meta-analysis shows that periodontal disease may be a potential risk factor for women suffering from breast cancer [139]. Many oral microbial metrics were strongly associated with breast cancer and nonmalignant breast diseases [140]. It is proved that *F. nucleatum* can bind to Fap2 in breast cancer tissue in a dependent manner, inhibit the accumulation of tumor-infiltrating T cells, and promote the growth and metastasis of tumors, which can be offset by an antibiotic treatment. Therefore, targeting *F. nucleatum* or Fap2 may be beneficial in the clinical treatment of breast cancer [141]. The further exploration of the effect of bacteria on breast cancer will provide new ideas for the diagnosis and treatment of breast cancer.

## 5. Conclusions

More and more evidence has indicated that there is a close relationship between the oral microbiota and tumor occurrence including in oral cancer, gastric cancer, colorectal cancer, pancreatic cancer, lung cancer, and breast cancer. There are three possible mechanisms of action of the oral microbiota in the pathogenesis of tumors [23]. Firstly, oral microbiota dysbiosis leads to chronic inflammation in which the inflammatory mediators that are produced induce or promote cell proliferation, mutation, oncogene activation, and angiogenesis [142]. For example, *oral streptococci* may be involved in the formation of certain reactive oxygen species (ROS), which induce apoptosis and tissue damage by damaging the nucleic acids, proteins, and lipids [143]. Secondly, oral microbiota dysbiosis may affect the metabolic pathways of host cells and promote tumorigenesis by affecting cell proliferation, cytoskeletal rearrangement, NF-κB activation, and the inhibition of apoptosis [144]. Thirdly, oral microbiota dysbiosis produces some oncogenic substances (e.g., the bacteria convert ethanol to acetaldehyde—a recognized carcinogen) and induces the malignant transformation of cells [145].

Certain oral microbial species can be observed in the saliva, subgingival plaque, mucosal tissue, tumor surface and intratumoral tissue, thus interring that oral microbiota may reach distal sites through different pathways to promote the tumorigenic process. Hematogenous spread is a possible acute incision route of oral pathogens during tooth extraction or trauma, particularly in patients with periodontitis who have high levels of periodontal pathogens, as transient bacteremia accompanies routine oral hygiene practices [128]. In addition, the osseo-enamel junction may also be a pathway for microbial invasion into chronic oral hygiene. Park Do-Young et al. suggested that the gingival sulcus (GS) and the junctional epithelium (JE) are the weakest point for microbes to invade the human body, where oral pathogens can settle for life and infiltrate the blood vessels and circulate throughout the body [146]. Gastrointestinal spread is another important route. The ecological invasion of the gut by the oral microbiota has a significant impact on digestive health. Among these, *P. gingivalis* has been shown to cause the dysbiosis of the intestinal microbiota, impairing the mucosal barrier function and leading to the spread of intestinal bacteria to the liver [108]. Overall, since the oral bacteria are actively participating in the induction of gut microbiota dysbiosis and tumor proliferation, targeting the microbiota to modulate the immune response reveals new avenues for cancer immunotherapy.

Interestingly, with the maturation of fecal transplantation techniques and their clinical application, a small number of dental researchers have hypothetically proposed oral microbiota transplantation (OMT). OMT may represent a cost-effective approach and can better reach the hard-to-reach, high-risk populations. However, based on the current state of knowledge, clinical recommendations for the use of OMT cannot be provided at this time. It is essential to better understand the retention of the transplanted oral biofilm while maintaining the natural balance between the resident oral microbiota and the host’s immune response [17].

At present, the early diagnosis of tumors, especially the early diagnosis of asymptomatic tumors, is receiving increasing attention. Traditional tumor detection methods all have various limitations, making it difficult to perform the large-scale screening of early-stage tumors [147]. Therefore, the early screening of susceptible people through non-invasive methods has become a hot topic in current tumor diagnosis research. As a valuable, non-invasive and easy-to-collect diagnostic tool, saliva holds great promise for biomarker research and development, health and disease monitoring and personalized medicine [148]. Salivary exosomes can transport tumor-specific contents to different parts of the body, including the salivary glands, resulting in the presence of disease-recognition markers in the saliva [149]. Zhou et al. reported the results of saliva testing in 47 patients with oral squamous cell carcinoma and showed that the average diagnostic accuracy based on oral microbiota was over 90% [55]. Huang’s cohort study [66] and Sun’s cohort study [69] both demonstrated the accuracy and sensitivity of using saliva microbiota to screen for GC (AUC of 91% and 97%, respectively). These studies suggest that the detection of oral microbiota is conducive to more accurate large-scale, low-cost screening of early-stage tumors and has promising applications [150,151].

Is it time to pay attention to the oral microbiota with insight into its therapeutic effect for cancer therapy? Oral microbiomes should continue to be explored in future trials to establish the scientific and clinical basis for tumor prevention and amelioration. To fully understand the relationship between pathogenic microorganisms and disease, future experiments could focus on the mechanisms of pathogen involvement in tumorigenesis by testing the response of human cells to living or heat-killed bacteria, or even to purified bacteriocin at the molecular level. Advances in the identification of biomarkers to personalize the treatment based on the patient’s unique microbiota and immunity profile will advance the treatment of tumors. A new era of oral microbiome research will benefit cancer patients.

## Figures and Tables

**Figure 1 microorganisms-10-02206-f001:**
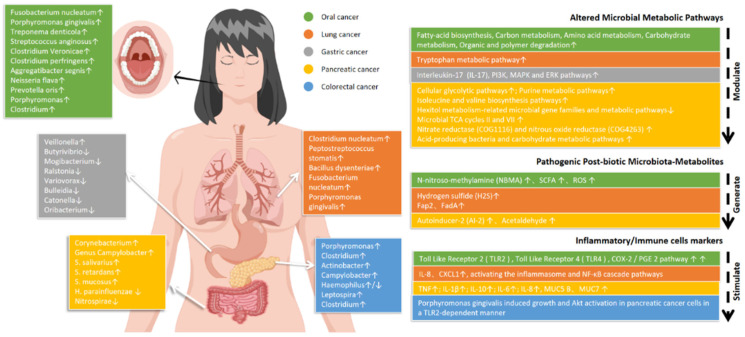
Oral Microbiota; Microbiota—metabolites and inflammatory markers in tumors of various sites. Locations are denoted by the following color code. IL: interleukin; SCFA: short-chain fatty acids; ROS: reactive oxygen species; TNF: tumor necrosis factor; CXCL: C-X-C motif chemokine; ↓: decreased level; ↑: increased level. Oral microbiota may be transmitted distally through the bloodstream and digestive tract. Figure 1 shows the changes in oral microbiota, microbial metabolic pathways, microbial metabolites, and inflammation and immune factors when the human body suffers from different kinds of tumors (oral cancer, gastric cancer, colorectal cancer, pancreatic cancer, and lung cancer).

**Table 1 microorganisms-10-02206-t001:** Oral organisms associated with oral cancer.

Subject	Organisms (Oral Bacteria)	Sample Type	Reference
47 OSCC patients and 48 healthy individuals as controls.	The proportions of *Actinobacteria*, *Fusobacterium*, *Moraxella*, *Bacillus*, and *Veillonella* species were higher in the disease group than they were in the control group.	saliva, subgingival plaque, the tumor surface, tumor tissue samples	[55]
60 OSCC patients and 120 gender and age-matched controls.	The proportions of *Prevotella oris, Neisseria flava, Neisseria* *flavescens/subflava, F. nucleatum, Aggregatibacter segnis, Streptococcus mitis,* and *Fusobacterium periodontium* were higher in the disease group than they were in the control group.	fresh OSCC biopsies samples	[56]
48 OSCC patients and 46 controls.	The proportions of *Prevotella, Campylobacter, Capnocytophaga,* *Solobacteria, Peptostreptococcus,* and *Catonella* were higher in the disease group than they were in the control group.	whole mouth fluid (WMF) and swab samples	[53]
Oral cancer patients (n = 50) and healthy subjects (n = 50).	The proportions of *Staphylococcus* and *Rothia* were higher in the disease group than they were in the control group.	swab samples	[54]
25 patients with OSCC and 24 healthy controls were recruited from Dr. B. Borooah Cancer Institute (BBCI), Guwahati, Assam, India.	The proportions of *P. melaninogenica, Streptococcus anginosus*, *Veillonella parvula, Prevotella pallens, Porphyromonas endodontalis, Prevotella nanceiensis, Dialister sp., Campylobacter ureolyticus*, *Fusobacterium sp., P. nigrescens, Neisseria bacilliformis,* and *Peptostreptococcus anaerobius* were higher in the disease group than they were in the control group.	samples of the whole saliva	[57]
Patients presenting with OLK (n = 36, average age: 60.6).	The proportions of *Fusobacterium, Leptotrichia, Campylobacter,* and *Rothia* were higher in the disease group than they were in the control group.	swabs	[51]
43 oral lichen planus patients and 21 mucosal healthy volunteers.	The proportions of *Fusobacterium, Leptotrichia,* and *Lautropia* were higher in the disease group than they were in the control group.	buccal scraping samples	[52]

**Table 2 microorganisms-10-02206-t002:** Oral organisms associated with gastric cancer (GC).

Subject	Organisms (Oral Bacteria)	Sample Type	Reference
293 patients included superficial gastritis (SG; n = 101), atrophic gastritis (AG; n = 93), and gastric cancer (GC; n = 99).	The proportions of presumed proinflammatory taxa, including *Corynebacterium* and *Streptococcus* were higher in the disease group than they were in the control group.	saliva sample	[66]
81 cases including SG, AG, intestinal metaplasia (IM) and GC from Xi’an, China.	Oral bacteria such as *Peptostreptococcus stomatis, Streptococcus anginosus, Parvimonas micra, Slackia exigua* and *Dialister* *pneumosintes* were enriched in cancerous tissues.	gastric mucosal samples	[67]
62 GC patients who underwent subtotal gastrectomy at The First Hospital of China Medical University.	Oral bacteria such as *Fusobacterium, Streptococcus*, *Peptostreptococcus,* and *Prevotella* were enriched in cancerous tissues.	gastric tissue samples	[68]
37 individuals with GC and 13 controls.	The proportions of *Veillonella, Prevotella, Aggregatibacter,* and *Megasphaera* increased were higher in the disease group than they were in the control group, while the proportions of *Leptotrichia, Rothia, Capnocytophaga, Campylobacter, Tannerella* and *Granulicatella* were lower.	saliva and plaque samples	[69]
57 newly diagnosed gastric adenocarcinomas and 80 healthy controls.	The proportion of *Firmicutes* was higher in the disease group than it was in the control group, while the proportion of *Bacteroidetes* was lower.	tongue coating sample	[70]
78 gastritis patients and 50 healthy individuals.	The proportion of *Campylobacter concisus* was higher in the disease group than it was in the control group.	tongue-coating samples	[71]
165 GC cases and 323 matched controls from Asian, African American, and European American populations.	The proportions of Neisseria mucosa and *Prevotella pleuritidis* were higher in the disease group than they were in the control group, while the proportions of *Mycoplasma orale* and *Eubacterium yurii* were lower.	pre-diagnostic buccal samples	[72]
12 GC cases and 20 matched controls (functional dyspepsia) in Singapore and Malaysia.	Oral bacteria such as *Lactococcus, Veilonella,* and *Fusobacteriaceae (Fusobacterium* and *Leptotrichia)* were enriched in cancerous tissues.	antral gastric biopsies	[73]
47 patients including SG, AG, gastric intraepithelial neoplasia (GIN), and GC.	Oral bacteria such as *Slackia, Selenomonas, Bergeyella* and *Capnocytophaga* were enriched in cancerous tissues.	gastric mucosal specimens	[65]

**Table 3 microorganisms-10-02206-t003:** Oral organisms associated with colorectal cancer (CRC).

Subject	Organisms (Oral Bacteria)	Sample Type	Reference
1165 cases with CRC and 739 cases for the periodontal bone loss.	Oral bacteria such as *Fusobacteria* were enriched in stool samples of CRC patients.	tissue sample	[92]
Matching samples of unstimulated saliva, cancer tissues or biopsies and stools were collected from 30 CRC and 30 HC patients.	The proportion of Salivary *Firmicutes*-to-*Bacteroides* ratio was higher in the disease group than it was in the control group.	unstimulated saliva, cancer tissues, or biopsies and stool samples	[93]
Individuals with either CRC (n = 99), colorectal polyps (n = 32) or healthy individuals as controls (n = 103).	Oral bacteria such as *Fusobacterium, Peptostreptococcus*, *Porphyromonas,* and *Micromonas* were enriched in the stool of patients with CRC or adenomas.	oral swabs, colonic mucosa, and stool samples	[94]
252 healthy and advanced CRC subjects.	Oral bacteria such as *F. nucleatum, Peptostreptococcus* *stomatis, Gemella morbillorum,* and *Parvimonas micra* were enriched in fecal samples of CRC patients.	fecal sample	[95]
Saliva samples from 14 CRC patients were collected.	The proportion of *F. nucleatum* was higher in the disease group than it was in the control group.	saliva samples	[96]
Individuals including 231 incident CRC cases and 462 controls.	The proportion of *Bifidobacteriaceae* was higher in the disease group than it was in the control group.	mouth rinse samples	[97]
Populations including CRC (99 subjects), colorectal polyps (32), or controls (103).	The proportions of *Haemophilus, Micromonas, Prevotella*, *Heterobacterium, Anaerobic, Neisseria,* and *Streptococcus* were lower in the disease group than they were in the control group.	oral swabs, colonic mucosal and stool samples	[98]
Mucosal samples from 59 patients undergoing surgery for CRC, 21 individuals with polyps and 56 healthy controls.	*Bacteroidetes Cluster 1* and *Firmicutes Cluster 1* were reduced in fecal samples of CRC patients, whereas *Bacteroidetes Cluster 2, Firmicutes Cluster 2, Pathogen Cluster* and *Prevotella Cluster* were enriched in them.	fecal and mucosal samples	[99]
Fecal microbiota in patients with adenomas (n = 233) and those without adenomas (n = 547) were analyzed.	Pro-inflammatory bacteria of the genera *Biliophilus*, *Desulfovibrio, Mogibacterium* and *various Bacteroidetes* were enriched in fecal samples of CRC patients.	fecal sample	[100]

**Table 4 microorganisms-10-02206-t004:** Oral organisms associated with pancreatic cancer.

Scheme	Organisms (Oral Bacteria)	Sample Type	Reference
30 PHC patients and 25 healthy controls.	The proportions of *Firmicutes, Fusobacteria,* and *Actinobacteria* were higher in the disease group than they were in the control group.	tongue coat samples	[121]
361 incident adenocarcinoma of pancreas and 371 matched controls.	The proportions of *P. gingivalis* and *A. actinomycetemcomitans* were higher in the disease group than they were in the control group, while the proportion of *Leptotrichia* was lower.	oral wash samples	[39]
Forty newly diagnosed PDAC patients, 39 IPMN patients, and 58 controls.	The proportions of *Firmicutes* and related taxa (*Bacilli, Lactobacillales, Streptococcaceae, Streptococcus,* and *Streptococcus thermophilus*) were higher in the disease group than they were in the control group.	saliva samples	[122]
405 pancreatic cancer cases and 416 matched controls.	The proportion of antibodies against *P. gingivalis ATTC 53978* was higher in the disease group than it was in the control group.	blood samples	[123]
10 resectable patients with pancreatic cancer and 10 matched healthy controls.	The proportions of *N. elongata* and *S. mitis* were lower in the disease group than they were in the control group, while the proportion of *G. adiacens* was higher.	saliva samples	[124]
patients with pancreatic cancer (n = 41) and healthy individuals (n = 69).	The proportions of *Lactobacillales, Bacilli, Streptococcus, Firmicutes*, *Actinomyces, Rothia, Leptotrichia, Lactobacillus, E. coli*, and *Enterobacteriales* were higher in the disease group than they were in the control group.	saliva samples	[120]
patients with suspected PCN (n = 105).	The proportion of *F. nucleatum* genome was higher in the disease group than it was in the control group.	cyst fluid and peripheral blood liquid biopsies	[118]
IPMN pancreatic cystic tumor cases and controls.	The reactivity of salivary IgA to *F. nucleatum* and the Fap2 mimotope increased.	paired plasma and saliva samples	[125]

**Table 5 microorganisms-10-02206-t005:** Oral organisms associated with respiratory tumors (lung cancer).

Subject	Organisms (Oral Bacteria)	Sample Type	Reference
Lung adenocarcinoma patients who did not smoke (cancer, n = 91) and healthy controls (control, n = 91).	The proportion of *Eillonella* was higher in the disease group than it was in the control group, while the proportions of *Mogibacterium*, *Butyrivibrio, Variovorax, Ralstonia, Catonella, Bulleidia*, and *Oribacterium* were lower.	saliva	[133]
Cases were subjects who were diagnosed with incident lung cancer (n = 114) and with controls (n = 114).	The proportions of *Bacilli* class and *Lactobacillales* order were higher in the disease group than they were in the control group, while the proportions of *Spirochaetia* and *Bacteroidetes* were lower.	mouth rinse samples	[134]
148 subjects with lung nodules from the NYU Lung Cancer Biomarker Center.	Oral bacteria such as *Veillonella, Streptococcus, Prevotella,* and *Haemophilus* were enriched in LC.	lower airway brushes	[136]

## Data Availability

Not applicable.

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
