# Peer review of "Oral Microbiota and Tumor—A New Perspective of Tumor Pathogenesis"

_microorganisms, 2022, doi:10.3390/microorganisms10112206_

Round 1
Reviewer 1 Report
Li et al. submitted a manuscript entitled "Oral microbiota and tumor - a new perspective of tumor pathogenesis". The authors discuss the recent advances for the oral microbiota in tumor developments. It is useful to summarize the current knowledge and helpful for people to better understand to latest progress in this field. However, several concerns are needed to be addressed.
Line 35-36 It would be better to provide some examples for which diseases these oral pathogens cause.
Line 54-57 Saliva microbiota is different from the microbiota in oral anatomic sites such as subgingival/supragingival plaque, tongue, hard palate, or buccal mucosa, etc. It would be better to describe the other oral microbiota as well. Mouth wash is also widely used in oral microbiome study to discover diagnostic markers for diseases. Please describe the application of mouth wash microbiome as well.
Line 101-105 The authors stated that recover the dysbiosis and dysfunction of oral microbiota is a big issue, however, another issue is to define the normal status and distinguish the disease status from the normal status in diverse populations. The authors need to comment on this issue.
Line 110-115 The three sentences about bacteria and tumorigenesis are unclear and should be re-written.
Line 177-181 The mechanism of ROS formation seems to be associated with oral cancer as well as a variety of other diseases. This paragraph needs to be re-order to a more appropriate section.
Line 230-234 The discussion here is mainly about the gastric microbiota and gastric cancer rather than the oral microbiota and gastric cancer, which may confuse the readers.
Line 362-367 Another large study regarding the oral microbiome and lung cancer can be included in the discussion: Vogtmann E, et al. The oral microbiome and lung cancer risk: An analysis of 3 prospective cohort studies. J Natl Cancer Inst. 2022 Aug 5:djac149. doi: 10.1093/jnci/djac149. PMID: 35929779.
Line 436 Oropharyngeal microbiota is different from the general meaning of the oral microbiota. Please make sure that the statement here is correct.
Author Response
Response to Reviewer 1 Comments
Thank you for taking the time out of your busy schedule to review our manuscript. We carefully followed the reviewers' requests, answered each question, and revised the manuscript carefully, highlighting all changes in the manuscript. Because of these suggestions, the revised manuscript becomes better and readers can get more valuable information.
Our reply to the amendment is as follows:
Point 1: Line 35-36 It would be better to provide some examples for which diseases these oral pathogens cause.
Response 1: Line 35-36: An corresponding example has been added to illustrate diseases caused by oral pathogens. “Selenomonas may cause serious human disease including bacteremia[7], and is associated with asthma severity[8].”
Point 2: Line 54-57 Saliva microbiota is different from the microbiota in oral anatomic sites such as subgingival/supragingival plaque, tongue, hard palate, or buccal mucosa, etc. It would be better to describe the other oral microbiota as well. Mouth wash is also widely used in oral microbiome study to discover diagnostic markers for diseases. Please describe the application of mouth wash microbiome as well.
Response 2: Line 54-57: The corresponding information has been added. “The oral cavity has several distinct microbial habitats, including subgingival plaque, supragingival plaque, buccal mucosa, keratinized gingiva, saliva, hard palate, tongue and tonsil[12]. The previous study analyzed the tongue dorsum, hard palate and several different parts of the oral cavity, and saliva with or without rinsing in 20 healthy subjects and demonstrated that no significant differences were seen between the study participants[13].” “Likewise, oral wash can be used as a sample to analyze oral microbiota characteristics[16].”
Point 3: Line 101-105 The authors stated that recover the dysbiosis and dysfunction of oral microbiota is a big issue, however, another issue is to define the normal status and distinguish the disease status from the normal status in diverse populations. The authors need to comment on this issue.
Response 3: Line 101-105: Additional discussion on the normal and disease states of the oral microbiota has been added. “There is a distinctive predominant bacterial flora of the healthy oral cavity that is highly diverse and is different from that of oral disease. Aas, J ø rn A et al. detected 141 different representative bacterial taxa in the oral cavity of healthy people[10]. However, when these bacteria lose their colonization advantage, the oral microbiome transforms into a dysbiotic state, including PH imbalance and symbiotic biofilms dysbiosis, which may accelerate the pathological processes of numerous diseases, by serving as a storeroom for opportunistic pathogens[37].”
Point 4: Line 110-115 The three sentences about bacteria and tumorigenesis are unclear and should be re-written.
Response 4: Line 110-115: These three sentences have been rewritten accordingly. “Imbalance in the overall bacterial community (change of diversity metrics) is involved in the development of tumors, which is mainly based on chronic inflammation and immunosuppressive. The changes in the abundance of individual bacterial species, particularly Peptostreptococcus, Prevotella and Parvimonas, can also induce chronic inflammation.”
Point 5: Line 177-181 The mechanism of ROS formation seems to be associated with oral cancer as well as a variety of other diseases. This paragraph needs to be re-order to a more appropriate section.
Response 5: Line 177-181: This paragraph has been reordered to the 5th chapter section (discussion section).
Point 6: Line 230-234 The discussion here is mainly about the gastric microbiota and gastric cancer rather than the oral microbiota and gastric cancer, which may confuse the readers.
Response 6: Line 230-234: Additional explanations have been provided for this section. “Most microorganisms in the stomach are considered to come from the external environment, particularly the oral cavity. Oral microbiota can enter the stomach through saliva and cause pathological changes in the corresponding areas[86].”
Point 7: Line 362-367 Another large study regarding the oral microbiome and lung cancer can be included in the discussion: Vogtmann E, et al. The oral microbiome and lung cancer risk: An analysis of 3 prospective cohort studies. J Natl Cancer Inst. 2022 Aug 5:djac149. doi: 10.1093/jnci/djac149. PMID: 35929779.
Response 7: Line 362-367: This valuable large cohort study has been included in the discussion. “Another large prospective study on the oral microbiome and lung cancer also demonstrated that multiple oral microbial measures were associated with lung cancer risk, especially in squamous cell carcinoma and smokers[135].”
Point 8: Line 436 Oropharyngeal microbiota is different from the general meaning of the oral microbiota. Please make sure that the statement here is correct.
Response 8: Line 436: The wrong statement here has been corrected. Replace “Oropharyngeal microbiota” with “Oral microbiota”.
Lastly, we would like to express our gratitude to you again for your guidance and for reviewing our revised paper once again. We hope that with the help of your guidance we can complete an excellent manuscript and sincerely hope that our manuscript will be published in your journal.

Reviewer 2 Report
An interesting and well written review on the oral microbiota and it’s potential influence on cancer is presented by Li et al. The submission suffers from some minor errors in English throughout and would benefit from a comprehensive proofreading. The information contained within the review is up to date and convers an extensive range of cancers however there are some issues. The main mechanism of egress for oral microbes to various sites describe in the submission is via the digestive, which is true of course, however from the context of oral microbes there is also potential entry via the cementum enamel junction which can be accessible in a number of situations, in this context likely poor chronic oral hygiene rather than an acute entry point due to extraction or trauma. In an effort to widen the scope of the article the authors due neglect details on mechanism they refer to leaving the reader with numerous questions. It may be a better approach for the authors to narrow their remit and include more specifics those this lack of detail may be intentional.
Author Response
Response to Reviewer 2 Comments
Thank you for taking the time out of your busy schedule to review our manuscript. We carefully followed the reviewers' requests, answered each question, and revised the manuscript carefully, highlighting all changes in the manuscript. Because of these suggestions, the revised manuscript becomes better and readers can get more valuable information.
Our reply to the amendment is as follows:
Point 1: The submission suffers from some minor errors in English throughout and would benefit from a comprehensive proofreading.
Response 1: We have comprehensively proofread the manuscript for English errors to make the language more accurate.
Point 2: The main mechanism of egress for oral microbes to various sites describe in the submission is via the digestive, which is true of course, however from the context of oral microbes there is also potential entry via the cementum enamel junction which can be accessible in a number of situations, in this context likely poor chronic oral hygiene rather than an acute entry point due to extraction or trauma.
Response 2: The entry via the cementum enamel junction has been partially added to the discussion, which makes the main mechanisms of oral microbial entry to various sites more comprehensive. “In addition, the osseo-enamel junction may also be a pathway for microbial invasion into chronic oral hygiene. Park, Do-Young et al. suggested that the gingival sulcus (GS) and the junctional epithelium (JE) are the weakest leaky point for microbes to invade the human body, where oral pathogens can settle for life and infiltrate the blood vessels and circulate throughout the body[146].”
Point 3: In an effort to widen the scope of the article the authors due neglect details on mechanism they refer to leaving the reader with numerous questions. It may be a better approach for the authors to narrow their remit and include more specifics those this lack of detail may be intentional.
Response 3: There are abundant studies on the mechanisms related to oral microorganisms and tumorigenesis. Our manuscript focuses on summarizing these studies from a more systematic and comprehensive perspective to help the reader build a well-rounded logical system, rather than focusing on presenting complex mechanisms, so there is not too much of a generalized summary of mechanism studies.
Lastly, we would like to express our gratitude to you again for your guidance and for reviewing our revised paper once again. We hope that with the help of your guidance we can complete an excellent manuscript and sincerely hope that our manuscript will be published in your journal.

Round 2
Reviewer 2 Report
The authors have taken on board most suggestions and the article is suitable for publication